# Application of the Iber Two-Dimensional Model to Recover the Water Quality in the Lurín River

**Omayra Luzmila Mori-Sánchez [1], Lia Ramos-Fernández [2],\*, Willy Eduardo Lluén-Chero [2], Edwin Pino-Vargas [3] and Lisveth Flores del Pino [4]**

1   Experimental Irrigation Area, Universidad Nacional Agraria La Molina, Lima 15024, Peru
2   Department of Water Resources, Universidad Nacional Agraria La Molina, Lima 15024, Peru
3   Department of Civil Engineering, Universidad Nacional Jorge Basadre Grohmann, Tacna 23000, Peru
4   Center for Research in Chemistry, Toxicology and Environmental Biotechnology, Universidad Nacional Agraria La Molina, Lima 15024, Peru
\*   Correspondence: liarf@lamolina.edu.pe

**Abstract:** The Lurín River is one of the main sources of water for the city of Lima. However, the discharge of domestic wastewater, the presence of dumps, and long periods of drought cause the deterioration of the water resource. In this study, *DO*, *BOD$_5$*, *E. coli*, *T*, *EC*, *TSS*, *U*, and *h* were monitored at 13 monitoring points spread over 20 km of river influence. This information was used to calibrate the parameters of $K_{dbo}$, $K_{aire}$, $K_{dos}$, and $K_{dec}$ in the Iber two-dimensional numerical model, obtaining values of 0.55 d$^{-1}$, [4.84 d$^{-1}$–80.65 d$^{-1}$], 10 g O$_2$ m$^{-2}$d$^{-1}$, and [1.49 d$^{-1}$–15.42 d$^{-1}$], respectively, with efficiencies ranging from "very good" to "satisfactory". In the hydraulic model, a discretization of the channel, banks, and plains of 3, 5, and 7 m, respectively, was considered, resulting in a computational calculation time of 4 days in each simulation. The greatest contamination occurs in July at km 5 + 400 up to the Pan-American bridge. Therefore, it is proposed to recover the river by optimizing the San Bartolo Wastewater Treatment Plant (WWTP) and a new WWTP in Pachacámac to avoid diffuse contamination, with discharge flows of 0.980 m$^3$s$^{-1}$ and 0.373 m$^3$s$^{-1}$, respectively, and 4 mg L$^{-1}$, 15 mg L$^{-1}$ and 1000 NMP/100 mL for *DO*, *BOD$_5$*, and *E. coli*, respectively.

**Keywords:** biochemical oxygen demand; *Escherichia coli*; Iber two-dimensional numerical model; dissolved oxygen

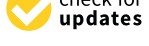



## 1. Introduction

The available water per capita in the world is reduced because of the increase in population, dumping of waste and chemicals from agricultural runoff, livestock, industrial and mining activities, and inadequate management of solid waste, which cause water shortages and deterioration of its quality. Contamination, without sufficient treatment, in rivers is a great concern, mainly because rivers are the source of water for domestic consumption; it is important to determine the quality of the water in rivers.

The Lurín river basin, which originates in the Andes Mountains, is located on the central coast of Peru, which has approximately two million inhabitants. Its climate is arid with scarce rainfall and inadequate management of water sources, leading to the inefficient development of agriculture and livestock, the main economic activities in the mentioned watershed [1]. Likewise, there is an accelerated process of unplanned urbanization, strong pressure for land and a risk of becoming a new industrial park. However, areas used for agriculture are still preserved, with tourism and archaeological potential resources, but with high poverty rates. In its lower section, the river is subject to important pressures derived from the extraction of water resources, the discharge of urban wastewater (treated and untreated), garbage disposal, livestock, and agricultural drainage. All this leads to an increasing scarcity of water and deterioration of its quality, negatively impacting aquatic ecosystems, and these factors are exacerbated during drought periods [2].

To solve this complicated problem, some tools are used to generate mathematical models of water quality to simulate working scenarios aimed at improving quality that can be used as a decision-making tool.

This study evaluated the current state of water quality using the two-dimensional Iber model, previously parameterized, to propose a recovery scenario for the Lurín River, to accomplish the environmental quality standard (ECAs) [3] and the Maximum Permissible Limit. (LMP) [4] of the Peruvian regulations.

## 2. Materials and Methods

### 2.1. Study Zone

The study was carried out in the lower part of the Lurín River (20 km), southeast of the city of Lima, located between 76°48′ and 76°54′ W longitude and 12°7′ and 12°16′ S latitude. This included the districts of Cieneguilla, Pachacamac, and Lurín from km 20 + 500 at the top of the Mototaxi bridge (L13) in the district of Cieneguilla, to km 0 + 578 at the top of the Panamericana South Bridge (L1) in the district of Lurín. Seven evaluations to obtain field information were performed between February and August 2019 (Figure 1).

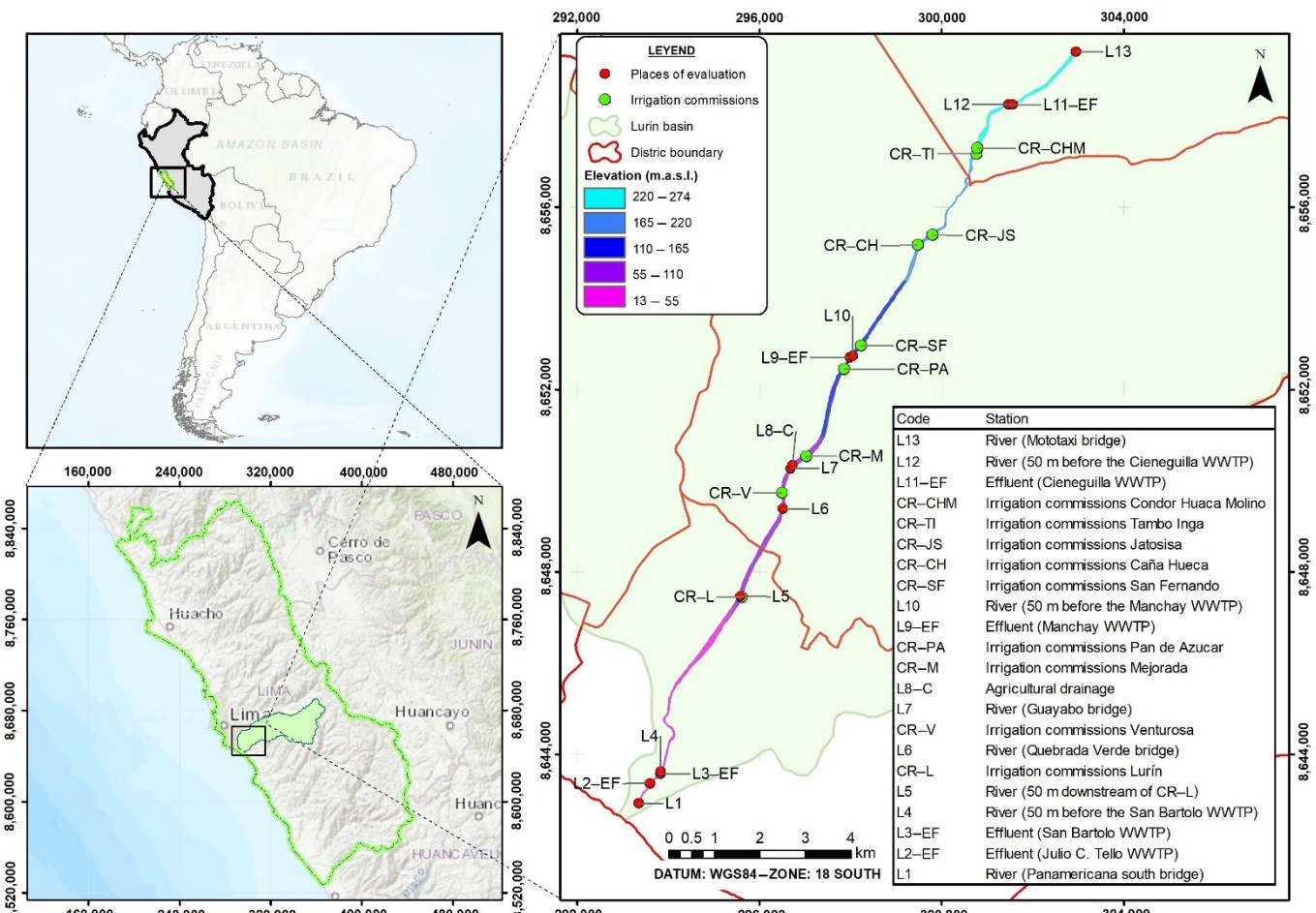

**Figure 1.** Geographical location of the evaluation stations and the intakes of the irrigation commissions in the lower part of the Lurín River.

### 2.2. Topographical Characterization

Images were captured with a DJI Zenmuse X4S camera attached to a Remotely Piloted Aircraft System (RPAS), Matrice 210 (DJI brand, Beijing, China), with an 80 m high flight plan, the lateral and frontal overlap of 75% and 70%, respectively, and a speed of 6 m s$^{-1}$. Further, the processing of images was done in the software photogrammetric Pix4Dmapper Pro (Pix4D SA, Prilly, Switzerland), and the topographical data was available for sections

starting in km 9 + 300 up to km 15 + 100, sections that corresponded to the irrigation commissions of Mejorada and Jatosisa, respectively.

The Digital Elevation Model (DEM) was generated in Civil 3D, in tiff format, with a resolution of 1.0 m. Due to the heterogeneity of the topography, it was partitioned with an unstructured mesh, which got a better fit to the surface: in the riverbed with 3 m; the banks with 5 m; and the plains with 7 m, as shown in Figure 2. Additionally, it is worth mentioning that the mesh size influences the computational time of each simulation, and four days were used to simulate the 20 km.

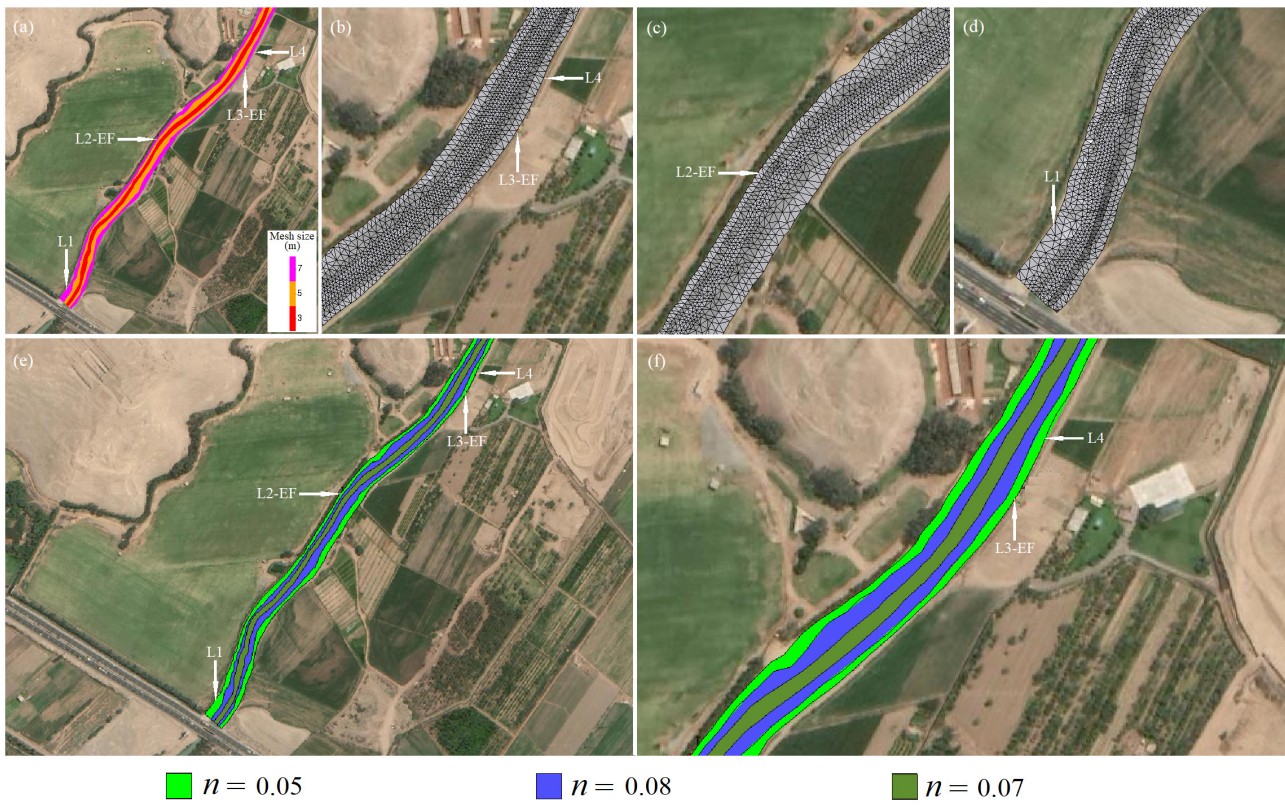

**Figure 2.** (**a**) Mesh size, (**b**) Mesh with DEM interpolation in L3-EF and L4, (**c**) Mesh with DEM interpolation in L2-EF, (**d**) Mesh with DEM interpolation in L1, (**e**) n in the lower part of the study section from L4 to L1, (**f**) n in L3-EF and L1. n is the is Manning's roughness coefficient.

### 2.3. Two-Dimensional Iber Model

The Iber model of water quality includes a hydrodynamic model, with basic information on the depth of the water, the speed, and the turbulent viscosity. All are necessary to solve the convection-diffusion equation for each polluting substance [5]. Bladé et al. [6] indicated that the hydrodynamic behavior of the river, in a two-dimensional model, is solved with two-dimensional St. Venant Equation (1), where the effects of turbulence [7] and surface friction by wind are incorporated:

$$\frac{\partial}{\partial t}(hU_x) + \frac{\partial}{\partial y}(hU_xU_y) + \frac{\partial}{\partial x}\left(hU_x^2 + g\frac{h^2}{2}\right) = -gh\frac{\partial Z_b}{\partial x} + \frac{\tau_{s,x}}{\rho} - \frac{\tau_{b,x}}{\rho} + \frac{\partial}{\partial x}\left(\nu_t h\frac{\partial U_x}{\partial x}\right) + \frac{\partial}{\partial y}\left(\nu_t h\frac{\partial U_x}{\partial y}\right)$$

$$\frac{\partial}{\partial t}(hU_y) + \frac{\partial}{\partial x}(hU_xU_y) + \frac{\partial}{\partial y}\left(hU_y^2 + g\frac{h^2}{2}\right) = -gh\frac{\partial Z_b}{\partial y} + \frac{\tau_{s,y}}{\rho} - \frac{\tau_{b,y}}{\rho} + \frac{\partial}{\partial x}\left(\nu_t h\frac{\partial U_y}{\partial x}\right) + \frac{\partial}{\partial y}\left(\nu_t h\frac{\partial U_y}{\partial y}\right)$$

(1)

where $h$ is the water depth, $U_x$, $U_y$ are horizontal velocities averaged in depth, $g$ is the acceleration due to gravity, $\rho$ is the water density, $Z_b$ is the depth to the bottom, $\tau_s$ is the friction on the open surface due to friction produced by the wind, $\tau_b$ is the friction due to the friction at the bottom, and $\nu_t$ is the turbulence kinematic viscosity.

The $\tau_b$ is evaluated by Manning's Equation (2):

$$\tau_{b,x} = \rho g h \frac{n^2 U_x |U|^2}{h^{\frac{4}{3}}} \quad \tau_{b,y} = \rho g h \frac{n^2 U_y |U|^2}{h^{\frac{4}{3}}} \tag{2}$$

where $n$ is Manning's roughness coefficient and $|U|$ is the resultant of the vectors $U_x$ and $U_y$.

Cea et al. [5] mentioned the spatial and temporal evolution of species, and pollutant variables are modeled with a generic convection-diffusion equation [Equation (3)]:

$$\frac{\partial}{\partial t}(hC) + \frac{\partial}{\partial x}(hU_x C) + \frac{\partial}{\partial y}(hU_y C) = \frac{\partial}{\partial x_j}\left( h\left( r_i + \frac{\nu_t}{S_{c,t}} \right) \frac{\partial C}{\partial x_j} \right) + S_c \tag{3}$$

where $C$ is the depth-averaged concentration of species, $S_c$ is a generic source term which depends on the species considered, $r_i$ is the coefficient of molecular diffusivity, $\nu_t$ is the turbulence kinematic viscosity, and $S_{c,t}$ (dimensionless) is the turbulence Schmidt number.

Cea et al. [5] indicated that the water quality module is completely paired with the hydrodynamic module. They share the same unstructured finite volume mesh adapted to topography, which allows defining parameters and visualization model outputs. Equation (3) is solved for each of the pollutants considered, where the reaction terms $S_c$ for each species are as follows:

1.  Temperature: four processes of heat transfer are considered, modeling the heat transfer between water and the atmosphere.

$$S_T = \frac{S_{Tem}}{C_\rho}(h) \tag{4}$$

where $S_T$ is the source term of contributions or sinks of heat (W m$^{-2}$), $S_{Tem}$ represents the heat transfer processes (1000 kg m$^{-3}$), and $C_\rho$ is the specific heat of water (4180 J kg$^{-1}$ °C$^{-1}$).

$$S_{Tem} = Q_{rad,in} + Q_{rad,out} + Q_{cond} + Q_{evap} \tag{5}$$

where $Q_{rad,in}$ is total net radiation absorbed by the water, $Q_{rad,out}$ is the long-wave radiation emitted by the water, $Q_{cond}$ is heat transferred by conduction, and $Q_{evap}$ is energy transferred by evaporation/condensation of the water.

1.  Dissolved Oxygen *(DO)* and Biological Oxygen Demand (*BOD*): Cea et al. [8] indicated that one of the main uses of *DO* in a body of water is the degradation of organic matter (*BOD*); see Equation (6).

$$BOD_5 = BOD_u \cdot \left( 1 - e^{-5 \cdot K} \right) \tag{6}$$

where $BOD_5$ is the *BOD* at 5 days, $BOD_u$ is the ultimate *BOD*, which occurs at the maximum possible oxygen consumption when the substrate has been completely degraded, and K is the organic matter degradation constant at river temperature, expressed in d$^{-1}$.

The source term for *DO* and *BOD* are shown in Equations (7) and (8), respectively:

$$S_{DO} = K_{aeration} * \theta_1^{(T-20)} * (DO_{sat} - DO) - K_{bod} \cdot \theta_2^{(T-20)} \cdot F_{oxc} \cdot BOD_u - \frac{K_{sod}}{h} \tag{7}$$

$$S_{BOD} = -K_{bod} \cdot \theta_2^{(T-20)} \cdot F_{oxc} \cdot BOD_u - \frac{V_{sBOD}}{h} * BOD_u \tag{8}$$

where $S_{DO}$ represents the reaeration source implemented in Iber, $S_{BOD}$ is a term that represents the reaction due to degradation or reaction with other substances present in the water, $K_{aeration}$ is the aeration constant at 20 °C, $T$ is the water temperature in °C, $DO_{sat}$ is the *DO* saturation concentration, $\theta_1$ is a correction coefficient for temperature ($\theta_1 = 1.024$), $K_{bod}$ is the carbonaceous organic matter degradation constant at 20 °C, $\theta_2$ is the temperature

correction coefficient ($\theta_2 = 1.047$), $F_{oxc}$ is a dimensionless attenuation factor due to low oxygen levels, $K_{sod}$ is the rate of demand of oxygen by the sediment in kg m$^{-2}$d$^{-1}$, $V_{sBOD}$ is the rate of sedimentation of organic matter in m, and $h$ is the depth of water in m.

$$K_{aeration} = K_{airh} + \frac{K_{airw}}{h} \tag{9}$$

where $K_{airh}$ is the reaeration constant at 20 °C based on the hydraulic characteristics of the river ($h$ and $U$) without considering the $V_{wind10}$, and is calculated by the Covar method [8] (see Table 1). $K_{airw}$ is the reaeration coefficient based on the $V_{wind10}$; it is calculated using Equation (10) proposed by Banks and Herrera [8]:

$$K_{airw} = 0.728V_{wind10}^{0,5} - 0.317V_{wind10} + 0.037V_{wind10}^2 \tag{10}$$

where $V_{wind10}$ is the wind speed measured at 10 m above the water level.

3. *E. coli*: a bacterium found in the gastrointestinal tract of homeothermic animals, such as humans, and therefore in urban wastewater.

$$S_d = -K_{dec} * C \tag{11}$$

where $S_d$ is the term that represents the bacterial disappearance, $K_{dec}$ is the constant of bacterial disappearance in $time^{-1}$, and $C$ is the concentration of *Escherichia coli* (*E. coli*), expressed in the most probable number per 100 mL (NMP/100 mL).

**Table 1.** Equations of the Covar method to estimate the reaeration constant $K_{aeration}$.

| Depth ($h$) in m<br>Water Speed ($U$)) in ms$^{-1}$ | Formula | Equation |
|---|---|---|
| If $h \leq 0.61$ m | Owens-Gibbs | $K_{airh} = 5.32\dfrac{U^{0.67}}{h^{1.85}}$ |
| If $h > 0.61$ and<br>$h > 3.45 * U^{2.5}$ | O'Connor-Dobbins | $K_{airh} = 3.93\dfrac{U^{0.5}}{h^{1.5}}$ |
| In other cases | Churchill | $K_{airh} = 5.026\dfrac{U}{h^{1.67}}$ |

SOURCE: Prepared with information from Cea et al. [8].

The degradation coefficient $K_{dec}$ is estimated with the Mancini [9] empirical formula, based on temperature, salinity, and solar radiation [10].

$$K_{dec} = (0.8 + 0.2Sal)1.07^{(T-20)} + 0.086\frac{I_0}{K_e H_c}\left(1 - e^{(-K_e H_c)}\right) \tag{12}$$

where $K_{dec}$ is the rate of disappearance averaged over a depth, $H_c$(d$^{-1}$), $Sal$ is the salinity (g L$^{-1}$), $T$ is the temperature of water (°C), $I_0$ is the incidence of solar radiation on the water surface (W m$^{-2}$), $K_e$ is the coefficient of extinction of light in water (m$^{-1}$), and $H_c$ is the depth of the vertical layer over which the *E. coli* spreads.

The rate of degradation of *E. coli* is largely dependent on the turbidity of the water, and is considered in the model through the light extinction coefficient $K_e$ [11]:

$$K_e = 2.619 + 0.129 * NTU \tag{13}$$

where $NTU$ is the turbidness of the water.

### 2.4. Hydrodynamic Characterization

Data on river flow were collected at eight points along the river and from four WWTP discharges (Cieneguilla, Manchay, San Bartolo, and Julio C. Tello); and in agricultural drainage. In addition, nine points were sampled in the streams of the irrigation commissions

and the Tinajas dry ravine, which becomes active during periods of rain with flows of about $0.025 \text{ m}^3\text{s}^{-1}$. At hydraulic sections of the river, mean speed and the height of the water were measured.

Manning's roughness coefficient was delineated from the Red-Green-Blue (RGB) images collected by the RPAS in the visits to the fields, by observing changes in land use and, therefore, variation in roughness. The values were selected according to [12].

Information from stations L13 and L1 was assigned to the conditions of input and output at the boundary model requirements, respectively. In this case, the drains and sources of the river were considered as river basins of the irrigation commissions and Wastewater Treatment Plants (WWTPs) discharges, respectively.

### 2.5. Characterization of Water Quality

Data on water quality were collected in the field at 13 evaluation points: eight points in the river; four in the WWTP discharges (Cieneguilla, Manchay, San Bartolo, and Julio C. Tello); and in agricultural drainage. $T$, Electrical Conductivity ($EC$), $DO$, $BOD_5$, $E.\ coli$, and Total Solids in Suspension ($TSS$) were measured. The first three parameters were measured in situ and the last three were measured in the laboratory, and the salt concentration ($Sal$) was estimated from $EC$ [13], see Equation (14):

$$Sal = -0.175 + 1.0053(EC) \tag{14}$$

Turbidness ($NTU$) was estimated from $TSSs$ [14], see Equation (15):

$$NTU = 1.0283 * TSS^{1.0282} \tag{15}$$

The information collected from these evaluations was compared to the Standards of Environmental Quality (ECA), for Category 3, to which the Lurín River belongs, where it is stated that concentrations for $BOD_5$ and $E.\ coli$ in the river should not be over $15 \text{ mg L}^{-1}$ and 1000 NMP/100 mL, respectively. In addition, $DO$ concentrations should not be less than $4 \text{ mg L}^{-1}$, and the Maximum Permissible Limit (LMP), indicated for $BOD_5$, is $100 \text{ mg L}^{-1}$. Uncontrolled discharges were identified in Pachacamac, near km 5 + 400. Similarly, higher contamination values were found on the final stretch of the river, from km 5 + 500 to km 0 + 578 where stations L5 and L1 were located, respectively. These stations were assigned as input and output boundaries under the conditions required by the model, respectively. In addition, concentrations of discharges from the WWTPs were included.

### 2.6. Climate Characterization

Hourly data (0 to 24 h) of total net radiation ($Ra_s$), wind speed ($V_{wind}$), air temperature ($T_{air}$), and relative humidity ($H_r$) were collected from the Alexander von Humboldt weather station. The albedo ($A_s$), which depends on the type of surface where the radiation falls and the angle of incidence ($\alpha$), was estimated with the expression $A_s = 1.18 \times \alpha^{-0.77}$, in which $\alpha$ was extracted from the webpage https://salidaypuestadelsol.com/sun/lima (accessed on 1 June 2021).

This information was used in Equation (5) to estimate the water temperature according to the dates of evaluation.

### 2.7. Model Calibration

Iber's two-dimensional model was used at a spatial scale of 1 m, the time in hours, and information for $DO$, $BOD_u$, $E.\ coli$, $EC$, $Q$, $Ra_s$, $H_r$, $V_{wind}$, and $T_{air}$.

The hydrodynamic model has been manually calibrated on 20 km of the river, with the Manning's roughness coefficients, after several trial and error simulations. Then, the simulated and observed $U$ and $h$ values were compared.

The water quality model was calibrated in the critical section from km 5 + 500 to km 0 + 578 (stations L5 to L1), with the parameters $K_{aeration}$, $K_{bod}$, $K_{sod}$, $K_{dec}$, and $V_{sBOD}$. It is important to remark that, from the information of $h$ and $U$ from the calibrated hydro-

dynamic model, $K_{aeration}$ was estimated using equations from Covar's method (Table 1). Values of $K_{bod}$ were obtained in a range of 0.02 to 3.4 $d^{-1}$, as recommended by Brown and Barnwell [15]. After several simulations and comparing concentrations of *DO* and $BOD_5$, observed and simulated, $K_{bod}$ was calibrated. For $K_{sod}$, a value of 10.0 g $O_2 m^{-2} d^{-1}$ was inserted according to the type of sludge of urban origin [16]. $K_{dec}$ was estimated according to Equation (12), and a zero value was given to $V_{sBOD}$, because turbulence and wind keep the particles in suspension.

The simulated concentrations were adjusted to those observed, and a trip back to the field with the data was organized to check for possible diffuse pollution, discharges, and/or undetected landfills. This is how a diffuse contamination was added at km 3 + 500 in Pachacamac.

After several simulations in March, May, and July, seven on each occasion, the model was calibrated, obtaining the Nash-Sutcliffe efficiency index (E), the ratio of the root of the mean quadratic error to the standard deviation of the variables of the observations (RSR), with limit values of E (−infinity to 1) and RSR (0 to +infinity), and rating the efficiency according to the scale proposed by Moriasi [17]. Furthermore, Pearson's correlation coefficient (R), which ranges from −1 to +1, was obtained with a *t*-Student test with a significance level (alpha) of 5%. The computational times for the hydrodynamic and water quality simulation were four days and one day, respectively. On the other hand, the use of the Iberplus plugin, which is an accelerated tool for flood modelling based on Iber, could be applied to reduce the computational time in the Iber water quality model [18,19].

### 2.8. Simulation of Scenarios for the Recovery of the River

Several simulations were performed for the critical month of July, including the optimization of the San Bartolo WWTP and the implementation of a WWTP in Pachacamac due to the discharge of uncontrolled domestic wastewater, with the result that the river complied with LMP [4] and ECA [3]. The procedure followed in the study is presented in a sequential diagram in Figure 3.

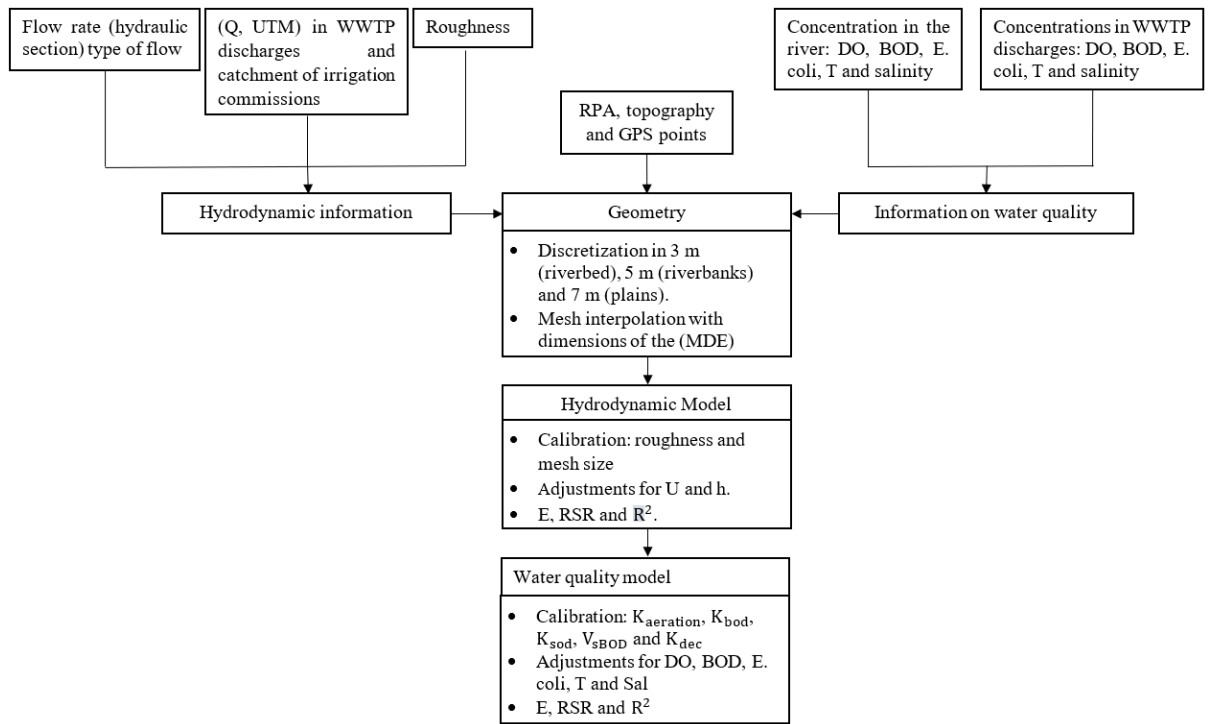

**Figure 3.** Flowchart of the methodology used in the study. WWTP is Waste Water Treatment Plant, RPAS is Remotely Piloted Aircraft System.

## 3. Results and Discussion

### 3.1. Hydraulic and Water Quality Characterization

The results of *U*, *h*, and *Q*, collected in the field at the evaluation points, indicated that *U* was less than 1.5 m s$^{-1}$ and the values of *h* were less than 0.8 m, with a Froude number of less than 1, that is, the flow of the river shows to be at a subcritical stage. The lowest *Q* was recorded in L5 and L4, caused by the irrigation commission catchment upstream of these points, with mean $\underline{x} \pm$ standard deviation S ($\underline{x} \pm$ S) of $3.22 \pm 4.92$ m$^3$s$^{-1}$ and $3.39 \pm 4.99$ m$^3$s$^{-1}$, respectively. In addition, the river was dry (*Q* = 0 m$^3$s$^{-1}$) in L5, on 19 June, 8 July, and 5 August 2019. There were high values of *Q* ($4.35 \pm 6.03$ m$^3$s$^{-1}$) in L13 because of the limited catchment from irrigation commissions upstream from this point. In contrast, L1 had the highest value of *Q* ($4.77 \pm 5.50$ m$^3$s$^{-1}$) caused by discharges from the San Bartolo WWTP (L3−EF) with values of $1.06 \pm 0.57$ m$^3$s$^{-1}$. The discharges to the river with the lowest *Q* were observed in L9-EF ($0.01 \pm 0.02$ m$^3$s$^{-1}$) and L8−C ($0.03 \pm 0.048$ m$^3$s$^{-1}$). The values ($\underline{x} \pm$ S) of *T*, *EC*, *TSS*, *DO*, *BOD$_5$*, and *E. coli*, are shown in Table 2, where the following can be noted:

- The *T* and *CE* comply with current Peruvian regulations.
- The *TSSs* have higher values ($114.49 \pm 159.31$ mg L$^{-1}$) at station L1 due to *TSSs* and higher *Q* value of the San Bartolo WWTP discharge. In contrast, station L13 shows the lowest values of *TSS* ($29.24 \pm 48.91$ mg L$^{-1}$) caused by a lower flow.
- In the river, from the Mototaxi bridge up to 50 m after the collect point of the Lurín irrigation commission (L13, L12, L10, L7, L6, and L5), the *DO* complies with ECA even in the dry period (lower flow), probably due to the photosynthetic activity of the algae present; for example, in the Quebrada Verde Bridge, and the Guayabo bridge (L6 and L7). In sections of the river from L4 to L1, there are uncontrolled discharges (diffuse contamination from Pachacamac) at km 5 + 400. In addition, discharge from San Bartolo WWTP (L3−EF) that does not meet the LMP causes a decrease in *DO*. This is more evident in the dry period and fails to comply with the ECA.
- The discharge of agricultural drainage (L8−C) with a high content of organic matter shows values of *BOD$_5$* ($97.23 \pm 139.75$ mg L$^{-1}$), which does not comply with the LMP on some dates. It causes an increase in *BOD$_5$* values ($22.06 \pm 22.52$ mg L$^{-1}$) downstream (L7) that does not comply with ECA. Similarly, the discharge from San Bartolo WWTP (L3−EF) does not meet the LMPs, due to an irregular process and lack of quality control.
- The San Bartolo WWTP (L3−EF) does not comply with LMP for all parameters and dates of evaluation, while the Julio C. Tello, Manchay, and Cieneguilla WWTPs did not comply with the LMP for *E. coli* in April, May, and July.
- The discharge of agricultural drainage (L8−C) has *E. coli* values of (271, $142.86 \pm 453$, 937.74 NMP/100 mL), which does not comply with LMP. This causes an increase in *E. coli* downstream at (L7) and does not comply with ECA. Similarly, discharge from San Bartolo WWTP (L3−EF) that does not comply with LMP causes failure to comply with the ECA for *E. coli* downstream at (L1).

### 3.2. Model Calibration Results

From the calibration of the model, the parameter $K_{aeration}$ revealed values in the range of (4.84 d$^{-1}$–80.65 d$^{-1}$). For $K_{bod}$, good efficiency was accomplished in the model with a value equal to 0.55 d$^{-1}$ at 20 °C. Figure 4a shows the values of E for DO and BOD$_5$ estimated with different values of $K_{bod}$, for seven simulations at each of the simulated dates (March, May, and July). The value of the parameter $K_{dec}$ was in the range of (1.49 d$^{-1}$–15.42 d$^{-1}$).

Figure 4b–f show the observed and simulated concentrations for DO, BOD$_5$, *E. coli*, T, and EC, as well as the efficiency indices E, RSR, and Pearson's correlation (R).

The efficiency of the model was rated from "very good" to "satisfactory", according to the E values (0.75–1.0, 0.65–0.75, 0.50–0.65, <0.5) and the RSR (0–0.5, 0.5–0.69, 0.6–0.7, <0.7) with a rating of "very good", "good", "satisfactory" and "unsatisfactory", respectively, a scale proposed by Moriasi et al. [17] with the t- Student test, for a significance level (alpha)

of 5%. Table 3 shows the E and RSR indices and Pearson's correlation (R) at the points of greatest river contamination.

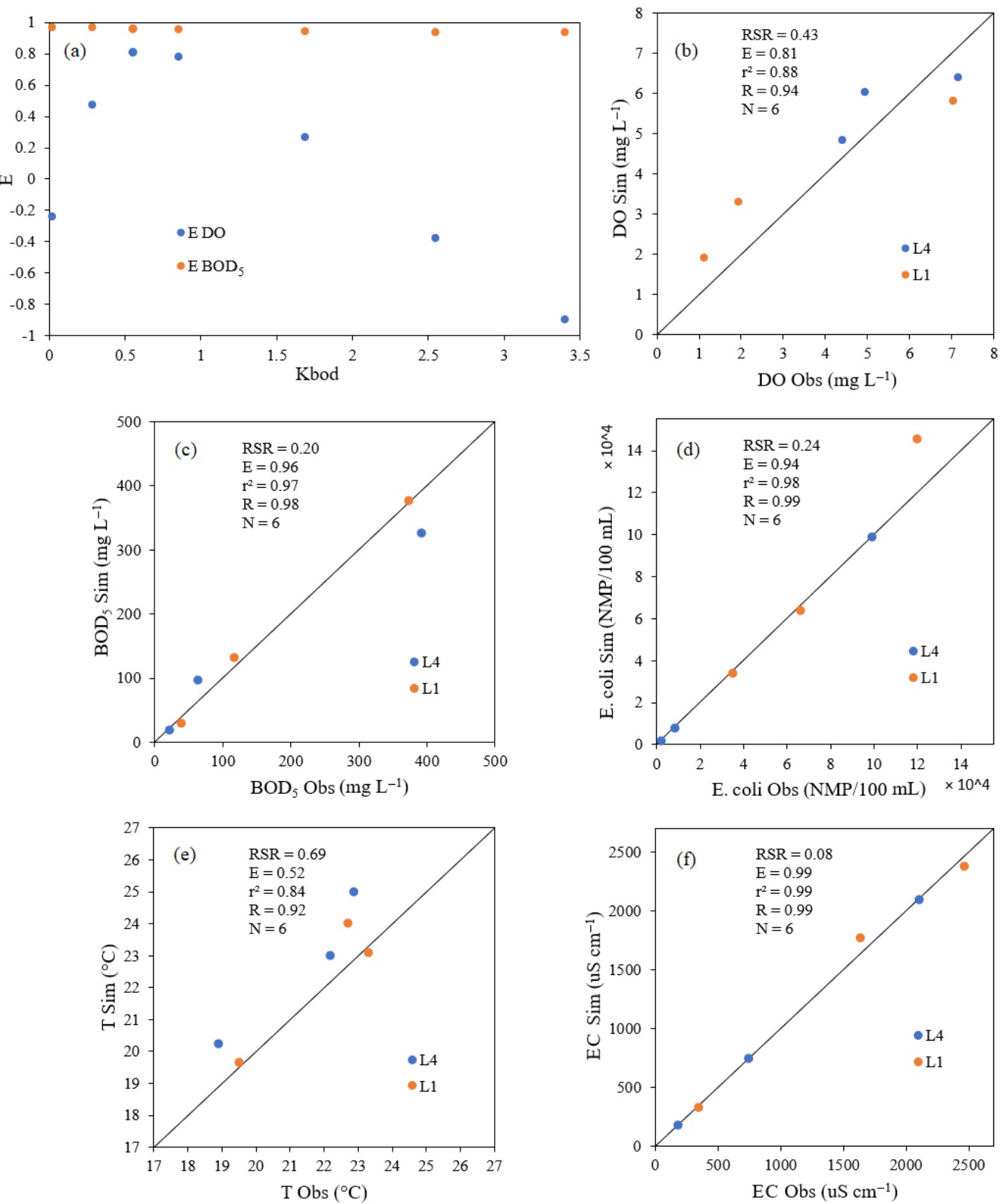

**Figure 4.** (**a**) Range of values of the E index due to changes in the parameter $K_{bod}$ in March, May, and July. Comparison between observed and simulated values according to: (**b**) dissolved oxygen, (**c**) $BOD_5$, (**d**) *E. coli*, (**e**) temperature, and (**f**) electrical conductivity.

**Table 2.** Means and standard deviation ($\underline{x}\pm$S) for water quality from data obtained in evaluations completed between February and August 2019.

| Stations | T (°C) | EC (uS cm$^{-1}$) | TSS (mg L$^{-1}$) | DO (mg L$^{-1}$) | BOD$_5$ (mg L$^{-1}$) | E. Coli (NMP/100 mL) |
|---|---|---|---|---|---|---|
| L13 | 23.31 ± 2.58 | 363.80 ± 195.90 | 29.24 ± 48.91 | 7.23 ± 0.97 | 1.93 ± 0.48 | 143 ± 378 |
| L12 | 24.37 ± 2.29 | 388.51 ± 222.50 | 27.9 ± 44.37 | 7.45 ± 1.61 | 4.22 ± 4.85 | 286 ± 488 |
| L11−EF | 25.00 ± 2.51 | 906.71 ± 156.05 | 3.70 ± 2.90 | 4.67 ± 0.35 | 2.83 ± 1.12 | 1714 ± 4536 |
| L10 | 23.51 ± 3.92 | 430.84 ± 244.42 | 44.58 ± 74.48 | 6.41 ± 0.93 | 8.37 ± 9.25 | 143 ± 378 |
| L9−EF | 25.10 ± 12.31 | 1588.50 ± 775.11 | 20.82 ± 13.57 | 6.29 ± 3.07 | 45.83 ± 30.85 | 27,000 ± 20,410 |
| L8−C | 22.44 ± 3.76 | 1264.83 ± 908.29 | 56.93 ± 75.55 | 6.23 ± 1.69 | 97.23 ± 139.75 | 271,143 ± 453,938 |
| L7 | 23.54 ± 3.24 | 757.49 ± 582.90 | 55.73 ± 84.51 | 5.75 ± 0.95 | 22.06 ± 22.52 | 60,714 ± 105,855 |
| L6 | 23.06 ± 3.29 | 571.39 ± 373.53 | 56.51 ± 93.64 | 8.73 ± 2.87 | 21.19 ± 14.92 | 6143 ± 12,090 |
| L5 | 24.78 ± 13.34 | 288.68 ± 192.83 | 163.92 ± 131.78 | 6.58 ± 3.54 | 7.97 ± 7.38 | 1500 ± 1069 |
| L4 | 20.73 ± 2.44 | 1080.44 ± 892.35 | 107.74 ± 165.59 | 5.48 ± 1.51 | 142.12 ± 147.77 | 23,857 ± 36,108 |
| L3−EF | 21.80 ± 11.08 | 2440.00 ± 1195.64 | 28.76 ± 18.24 | 0.83 ± 0.61 | 307.31 ± 198.08 | 101,800 ± 61,576 |
| L2−EF | 23.59 ± 3.25 | 1843.00 ± 156.38 | 12.50 ± 6.22 | 5.57 ± 0.65 | 41.81 ± 18.93 | 5000 ± 7234 |
| L1 | 21.27 ± 2.44 | 1549.43 ± 974.96 | 114.49 ± 159.31 | 3.77 ± 2.69 | 243.66 ± 225.86 | 72,428 ± 55,220 |

**Table 3.** Efficiency indices E and RSR, and Pearson's correlation (R) in the calibration of the Iber model for *T*, *EC*, *DO*, *BOD$_5$,* and *E. coli.*

| Parameters | L4: 50 m before WWTP San Bartolo | | | L1: South Pan−American Bridge | | | L4 and L1 | | |
|---|---|---|---|---|---|---|---|---|---|
| | E | RSR | R | E | RSR | R | E | RSR | R |
| DO | 0.546 | 0.674 | 0.805 | 0.806 | 0.440 | 0.974 | 0.813 | 0.433 | 0.940 |
| BOD$_5$ | 0.932 | 0.260 | 0.989 | 0.995 | 0.070 | 0.998 | 0.959 | 0.202 | 0.983 |
| E. coli | 0.999 | 0.005 | 0.999 | 0.823 | 0.421 | 0.994 | 0.944 | 0.237 | 0.989 |
| T | 0.250 | 0.866 | 0.965 | 0.790 | 0.458 | 0.941 | 0.518 | 0.690 | 0.917 |
| EC | 0.999 | 0.003 | 0.999 | 0.988 | 0.108 | 0.994 | 0.994 | 0.076 | 0.997 |

### 3.3. Simulation in Present Conditions

Figure 5 shows the longitudinal profile from km 5 + 500 (L5) to km 0 + 578 (L1), indicating the distances at which each discharge is found (July). In L5, the river flow is zero, initiating diffuse contamination in Pachacamac, 2164 m downstream of L5, with concentrations of *BOD$_5$* and *E. coli* of 341.68 mg L$^{-1}$, and 2.2 × 10$^5$ NMP/100 mL, respectively. At 2126 m downstream is the San Bartolo WWTP with discharges of *DO*, *BOD$_5$*, and *E. coli* of 0.65 mg L$^{-1}$, 469.95 mg L$^{-1}$ and 1.4 × 10$^5$ NMP/100 mL, respectively, and at 300 m the Julio C. Tello WWTP is found with discharges of *DO*, *BOD$_5$*, and *E. coli* of 5.67 mg L$^{-1}$, 35.29 mg L$^{-1}$ and 8 × 10$^3$ NMP/100 mL, respectively.

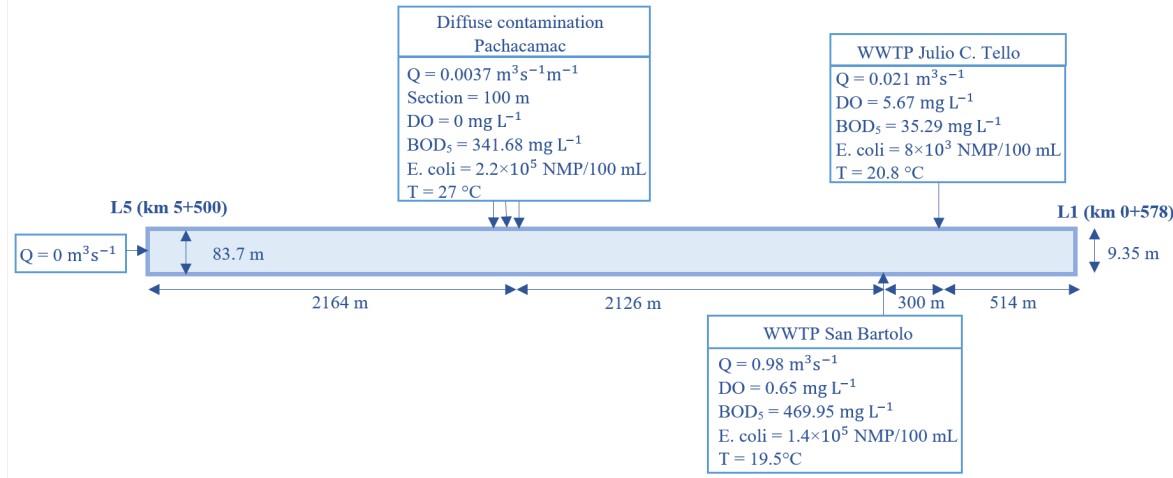

**Figure 5.** Schematic of the longitudinal profile from L5 to L1 with distances at which the discharges are found. Analysis date: July 2019.

Figure 6 shows the longitudinal profiles of the current simulation at the most critical condition time (July), where it is noted that the ECAs for $BOD_5$ and *E. coli* are not met in the section of the river from L5 to L1, because of the high concentrations of $BOD_5$ and *E. coli* from the diffuse contamination in Pachacamac. The *DO* does not comply with the ECAs between the San Bartolo WWTP and the Panamericana South Bridge because discharges of the San Bartolo WWTP with a water flow of 0.976m$^3$ s$^{-1}$, and concentrations of 0.65 mg L$^{-1}$, 469.95 mg L$^{-1}$, $1.4 \times 10^5$ NMP/100 mL of *DO*, $BOD_5$ and *E. coli*, respectively, do not comply with the LMP. In addition, the existence of solid waste landfills and other areas, such as informal latrines, were observed on the Panamericana south bridge.

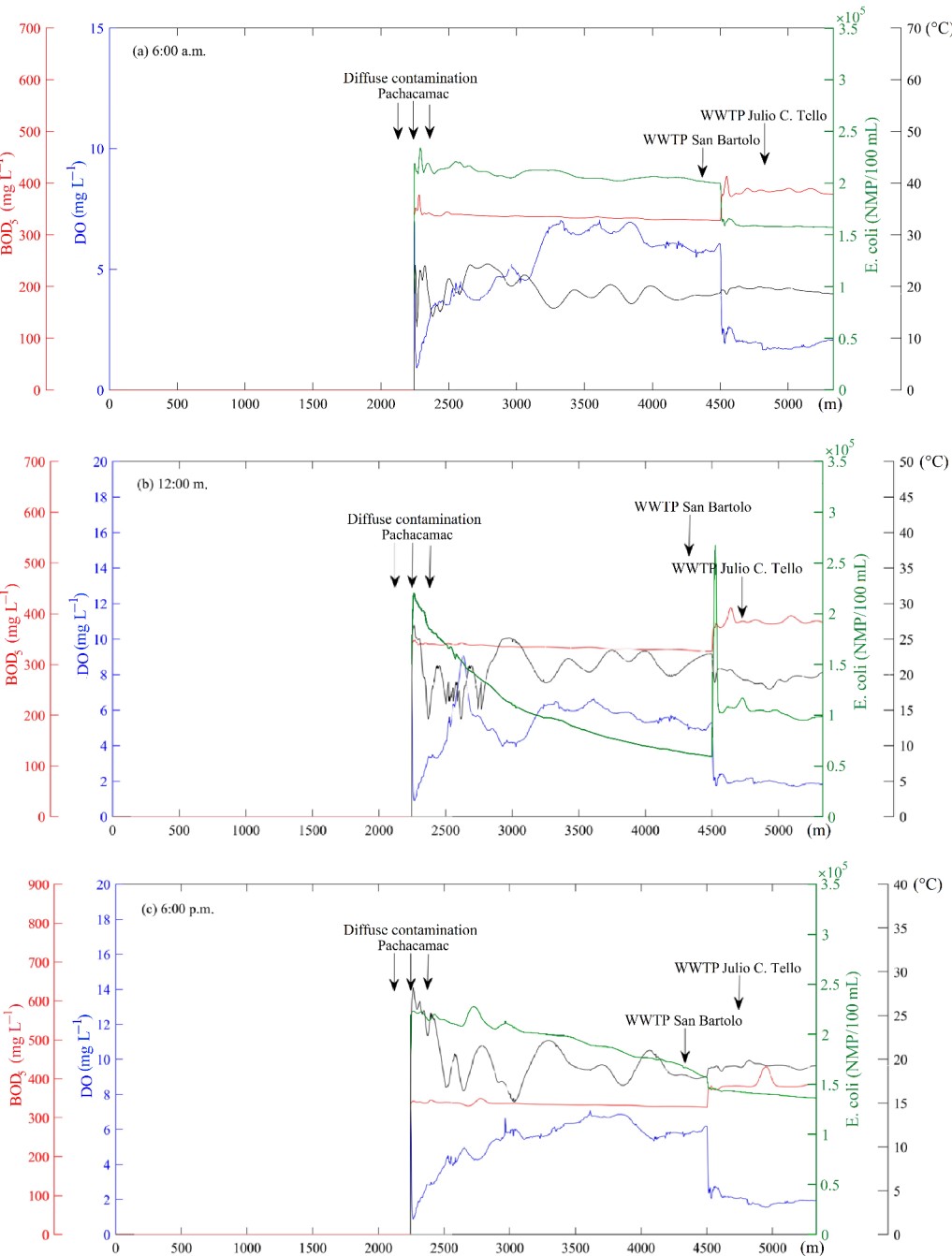

**Figure 6.** Longitudinal profile of the current situation of the river according to *DO*, $BOD_5$, *E. coli*, and *T* (from L5 to L1): 6 a.m. Table S1 (**a**), 12 m. Table S2 (**b**) and 6 p.m. Table S3 (**c**). Red line is $BOD_5$, blue line is *DO*, green line is *E. coli*, and black line is *T*.

### 3.4. Simulation of Recovery

The river recovery scenario was generated for July, with the improvement of the San Bartolo WWTP with a discharge flow of 0.98 m$^3$s$^{-1}$, and implementing the Pachacamac WWTP with a flow of 0.37 m$^3$s$^{-1}$, and both with concentrations of 4 mg L$^{-1}$, 15 mg L$^{-1}$ and 1000 NMP/100 mL of *DO*, *BOD$_5$*, and *E. coli*, respectively, which allows them to follow the LMP and ECA that are required for the river.

From the calibration of the hydrodynamic model, the discharge flow for the new WWTP was estimated. After several trial and error simulations, it was identified that for the month of the lowest flow (critical), the estimated flow of wastewater dumped into the river in Pachacamac was 0.0037 m$^3$s$^{-1}$m$^{-1}$, in a section of 100 m, with a total flow of 0.37 m$^3$s$^{-1}$. The discharges must comply with LMP and ECA for a river of this category, and for that several simulations were performed. Values of 4 mg L$^{-1}$, 15 mg L$^{-1}$ and 1000 NMP/100 mL were estimated for *DO*, *BOD$_5$*, and *E. coli*, respectively.

Figure 7 shows the longitudinal profiles of the polluting substances *DO*, *BOD$_5$*, *E. coli*, and *T*, for 6:00 a.m., 12:00 m., and 6:00 p.m. There, it is observed that after discharges take place in the new Pachacamac WWTP and San Bartolo WWTP, they continue for a segment of approximately 200 m in which ECAs are not met. However, this segment is considered a mixed one. It is also observed that as *T* increases, *DO* decreases, in an inverse relationship. Likewise, with increases in *BOD$_5$*, the *DO* decreases. Finally, *E. coli* at noon is reduced by 72%, because of higher solar radiation, with values up to 250 NMP/100 mL, as seen in Figure 7b, with higher values close to 1000 NMP/100 mL.

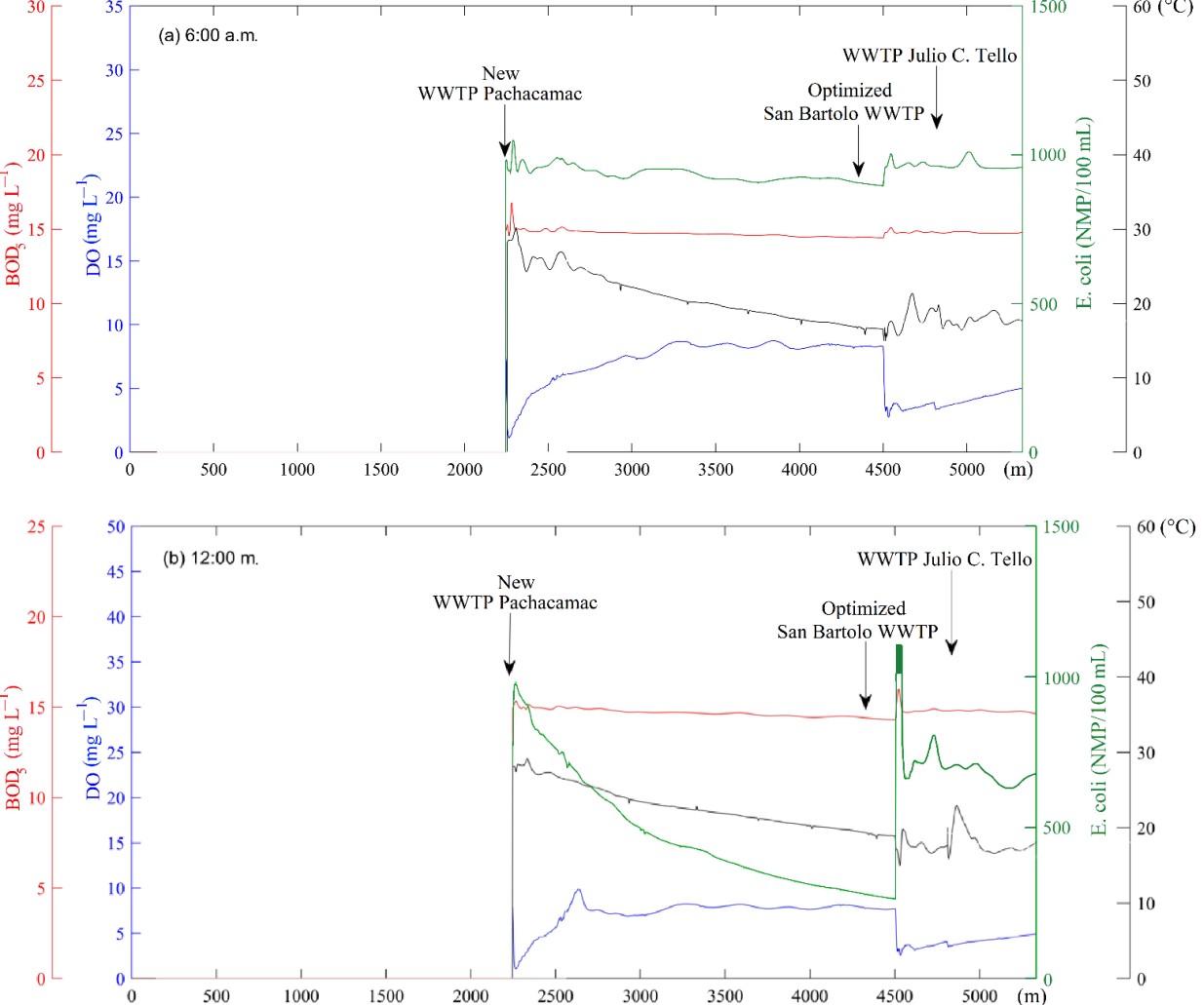

**Figure 7.** *Cont.*

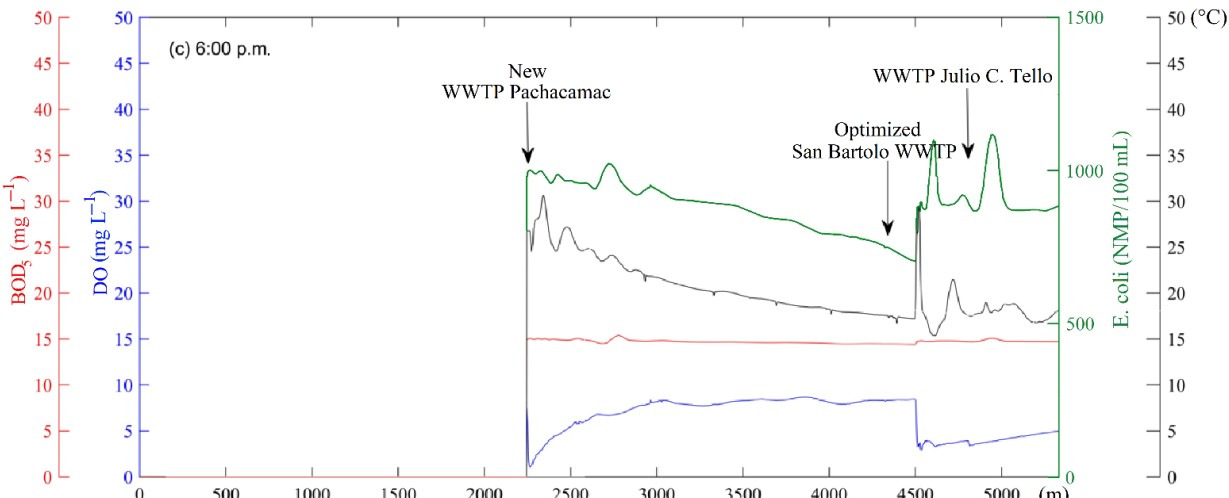

**Figure 7.** Longitudinal profile of river recovery according to *DO*, *BOD₅*, *E. coli,* and *T* (from L5 to L1): 6 a.m. (**a**), 12 m. (**b**) and 6 p.m. (**c**). Red line is $BOD_5$, blue line is *DO*, green line is *E. coli*, and black line is *T*.

To observe a two-dimensional behavior of the river, information was extracted from *h*, *U*, $U_x$, $U_y$, *DO*, $BOD_u$, *E. coli*, and *T*, in the cross section, 25 m after the San Bartolo WWTP (as shown in Figure 8), where it is noted that because of a greater influence of the discharges from San Bartolo WWTP, the difference in concentrations along the cross-sectional profile is high. It was also seen that on the right bank of the river, *U* reaches values of 0.27 ms$^{-1}$ and *h* to 0.09 m, which are lower with respect to the left bank, with values up to 0.49 ms$^{-1}$ and 0.28 m of *U* and *h*, respectively, due to the contribution of discharges from San Bartolo WWTP. The parameter *h* influences the concentration of contaminants because, at a lower depth, it will be completely reaerated, generating a higher concentration of oxygen. For that reason, *DO* and *h* show an inverse relationship. At noon, *T* reaches values of 21.5 °C; on the other hand, at a higher concentration of $BOD_u$, there is a greater amount of organic matter to degrade and, therefore, the *DO* is lower.

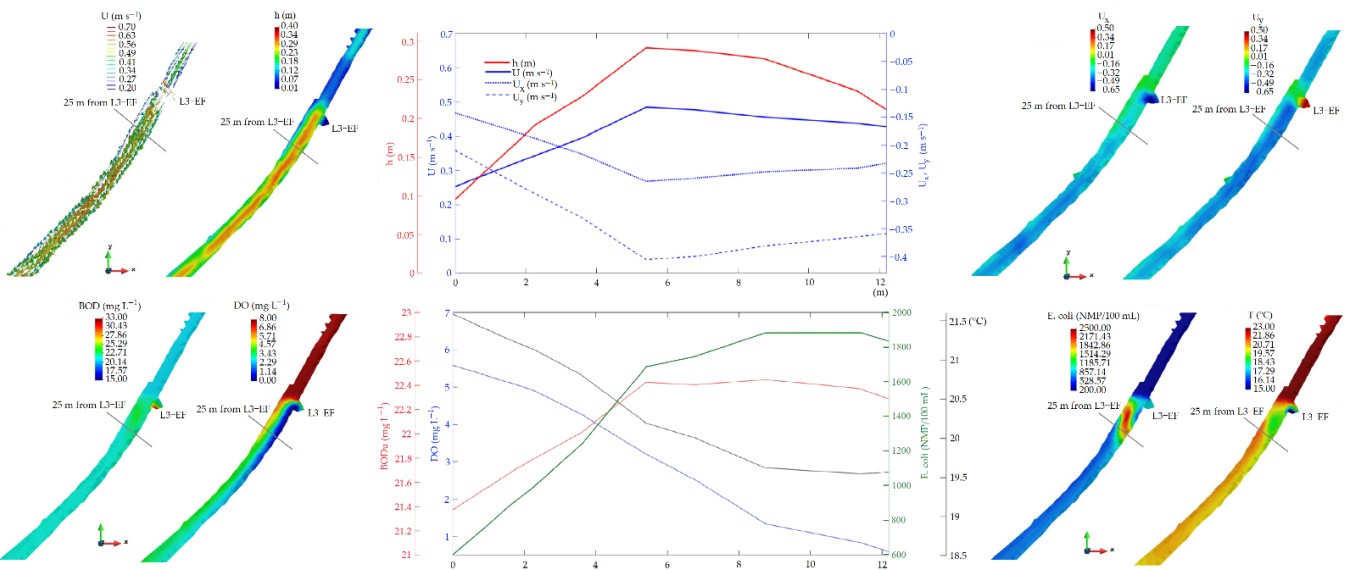

**Figure 8.** Cross-sectional profile of the recovery of the river according to *DO*, $BOD_u$, *E. coli*, *T*, *h*, *U*, $U_x$, and $U_y$, downstream San Bartolo WWTP, at 12:00 m. Top panel: red line is *h*, blue line is *U*, dotted blue line is $U_x$, dashed blue line is $U_y$. Bottom panel: red line is $BOD_u$, blue line is *DO*, green line is *E. coli*, black line is *T*.

In the cross sections, the highest concentration is found on the right side, due to the difference in speeds and heights, in addition to the great influence of the discharge from the San Bartolo WWTP, located at that end of the river.

## 4. Conclusions

The calibration of the Iber model revealed a performance ranging from "very good" to "satisfactory", with values of E, RSR, and $R^2$(0.813, 0.433, and 0.883) for *DO*, (0.959, 0.202, and 0.967) for *BOD$_5$*, (0.944, 0.237, and 0.979) for *E. coli*, and (0.518, 0.690, and 0.841) for *T*, with calibrated parameters of 0.55 d$^{-1}$, (4.84 *d$^{-1}$*–80.65 d$^{-1}$), 10 g O$_2$ m$^{-2}$d$^{-1}$, 0 m d$^{-1}$, and (1.49 d$^{-1}$–15.42 d$^{-1}$) for $K_{bod}$, $K_{aeration}$, $K_{sod}$, $V_{sBOD}$, and $K_{dec}$, respectively.

The most polluted area is located around the Panamericana south bridge, the critical month is July, with a flow of 1.2 m$^{-3}$ s$^{-1}$, and values of 1.12 mg L$^{-1}$, 372.69 mg L$^{-1}$, and 1.2 × 10$^5$ NMP/100 mL for *DO*, *BOD$_5$*, and *E. Coli*, respectively. This does not comply with the ECA category 3 due to (i) diffuse contamination in the Pachacamac district, and (ii) inadequate operation of the San Bartolo WWTP, which fails to comply with LMP on all evaluated dates and for all substances, with values of 0.44 mg L$^{-1}$, 469.95 mg L$^{-1}$, and 1.4 × 10$^5$ NMP/100 mL for *DO*, *BOD$_5$* and *E. Coli*, respectively. Diffuse contamination was estimated in Pachacamac with flows that go from 0.001 to 0.0037m$^{-3}$ s$^{-1}$ m$^{-1}$, and values of 0 mg L$^{-1}$, 888.37 mg L$^{-1}$ and 2.2 × 10$^5$ NMP/100 mL, for *DO*, *BOD$_5$* and *E. coli*, respectively.

It is proposed to recover the river by optimizing the San Bartolo WWTP and a new WWTP in Pachacámac to avoid diffuse contamination, with discharge flows for the critical month of July, of 0.980 m$^3$s$^{-1}$ and 0.373 m$^3$s$^{-1}$, respectively, with concentrations that meet the ECA category 3: 4 mg L$^{-1}$, 15 mg L$^{-1}$, and 1000 NMP/100 mL for *DO*, *BOD$_5$* and *E. coli*, respectively, with a flow of 0 m$^3$s$^{-1}$ before diffuse pollution and a flow of 0.209 m$^3$s$^{-1}$, 50 m before the San Bartolo WWTP.

**Supplementary Materials:** The following are available online at https://www.mdpi.com/article/10.3390/hydrology10040084/s1.

**Author Contributions:** Conceptualization, E.P.-V. and L.R.-F.; methodology, E.P.-V., L.R.-F., O.L.M.-S. and W.E.L.-C.; validation, E.P.-V., L.R.-F., L.F.d.P. and O.L.M.-S.; investigation, E.P.-V., L.R.-F., O.L.M.-S. and L.F.d.P.; resources, L.R.-F., L.F.d.P. and W.E.L.-C.; data curation, E.P.-V., L.R.-F. and O.L.M.-S.; writing—original draft preparation, O.L.M.-S., L.R.-F and W.E.L.-C.; writing—review and editing, L.R.-F., E.P.-V. and L.F.d.P.; supervision, E.P.-V., L.R.-F. and L.F.d.P. All authors have read and agreed to the published version of the manuscript.

**Funding:** This research was funded by The National Fund for Scientific and Technological Development (FONDECYT) of CONCYTEC-Peru, under the project "Surface and underground water resource management system for the safe use of water in climate change scenarios in the Lurín river basin" (Project No. 157-2017-FONDECYT-Peru).

**Institutional Review Board Statement:** Not applicable.

**Informed Consent Statement:** Not applicable.

**Data Availability Statement:** The data presented in this study are available on request from the corresponding author.

**Acknowledgments:** This paper was written as a result of collaboration that was initiated in the Project No. 157-2017-FONDECYT-Peru.

**Conflicts of Interest:** The authors declare no conflict of interest.

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
