# Peer review of "Application of the Iber Two-Dimensional Model to Recover the Water Quality in the Lurín River"

_hydrology, doi:10.3390/hydrology10040084_

Round 1

Reviewer 1 Report

The overly general introductory paragraph cites a paper written about India, which makes no sense for this manuscript, which is about a river in Peru.

The introduction needs to give a more comprehensive view of water quality in Peru and in the Lurin River, what the major uses of water from this River are, and how many people live in it and where the population is particularly concentrated (instead of stating that this basin is “one of the most demographically populated”)

How or why were these particular water quality parameters chosen for this study?  If there is significant mining in the watershed, then it would make a lot of sense to include pH and heavy metals to help understand the contribution of mining activities to water quality degradation…although electrical conductivity will help with this determination also.

The section of the paper (the first section 1.1) that includes discussion of the two-dimensional Iber Model is out of place and belongs in the methods section.

Numbering of sections in this paper needs to be fixed.

There needs to be a scale in Figure 1.

There must be labels for the numbers (UTM coordinates?) on the X and Y axes (or they should be eliminated and an inset map should be included to show where in Peru this study area is) as the significant proportion of geographically illiterate readers of this manuscript will not know what these numbers mean.

Page 6: There is no explanation of what “Standards of Environmental Quality (SEQ)” means. I assume it refers to national regulations in Peru, but as the audience is international this needs to be explained.

Section 1.4: Is there a good reason for the spatial scale of 1m and the time sale at (1) hour? 1m corresponds with the resolution of the digital elevation model but not with the meshes of 3, 5, and 7m that are noted in the Topographical Characterization.  It is unclear that there is an advantage to using a 1m spatial scale (compared to 3m or 5 m for example) in a study area of this size.

It is unclear why the focus on the month of July for “optimizing” the Pachacamac WWTP if the San Bartolo WWTP does not comply with any standards on any dates.

Author Response

Answer highlighted in green

1. The overly general introductory paragraph cites a paper written about India, which makes no sense for this manuscript, which is about a river in Peru.

We speak in general about the pollution in the world, not specifically about the Indian river, so the paper is not not include

2. The introduction needs to give a more comprehensive view of water quality in Peru and in the Lurin River, what the major uses of water from this River are, and how many people live in it and where the population is particularly concentrated (instead of stating that this basin is “one of the most demographically populated”

Modify in the text

3. How or why were these particular water quality parameters chosen for this study? If there is significant mining in the watershed, then it would make a lot of sense to include pH and heavy metals to help understand the contribution of mining activities to water quality degradation…although electrical conductivity will help with this determination also.

The mining activity is not relevant in this river. Modify In the text

4. The section of the paper (the first section 1.1) that includes discussion of the two-dimensional Iber Model is out of place and belongs in the methods section. Numbering of sections in this paper needs to be fixed.

Modify in the text

5. There needs to be a scale in Figure 1. There must be labels for the numbers (UTM coordinates?) on the X and Y axes (or they should be eliminated and an inset map should be included to show where in Peru this study area is) as the significant proportion of geographically illiterate readers of this manuscript will not know what these numbers mean.

We change figure 1

6. Page 6: There is no explanation of what “Standards of Environmental Quality (SEQ)” means. I assume it refers to national regulations in Peru, but as the audience is international this needs to be explained.

Modify in the text

7. Section 1.4: Is there a good reason for the spatial scale of 1m and the time sale at (1) hour? 1m corresponds with the resolution of the digital elevation model but not with the meshes of 3, 5, and 7m that are noted in the Topographical Characterization. It is unclear that there is an advantage to using a 1m spatial scale (compared to 3m or 5 m for example) in a study area of this size.

The mesh size was generated with a DEM of 1m resolution. It is important to mention that the mesh size influences the computational time of each simulation; if a 1m mesh had been used, the computational time would have been longer than 4 days. But, with the mixed mesh of 3, 5 and 7 m it was enough to characterize the results.

8. It is unclear why the focus on the month of July for “optimizing” the Pachacamac WWTP if the San Bartolo WWTP does not comply with any standards on any dates.

In July, the highest pollution of the river occurs in the L5 to L1 section. Thus,  it is proposed to recover the river by optimizing the San Bartolo WWTP and a new WWTP in Pachacámac to avoid diffuse contamination.

Note.- In the first version “implementation”  for us was considered a new WWTP.

Reviewer 2 Report

I think the paper is of quality

 I wish to recommend introducing the topic of connectivity

See here for some ideas

Keesstra, S., Nunes, J. P., Saco, P., Parsons, T., Poeppl, R., Masselink, R., & Cerdà, A. (2018). The way forward: can connectivity be useful to design better measuring and modelling schemes for water and sediment dynamics?. Science of the Total Environment, 644, 1557-1572.

López-Vicente, M., Kramer, H., & Keesstra, S. (2020). Effectiveness of soil erosion barriers to reduce sediment connectivity at small basin scale in a fire-affected forest. Journal of Environmental Management, 278, 111510.

Rodrigo Comino, J., Keesstra, S. D., & Cerdà, A. (2018). Connectivity assessment in Mediterranean vineyards using improved stock unearthing method, LiDAR and soil erosion field surveys. Earth Surface Processes and Landforms, 43(10), 2193-2206.

Author Response

The authors have reviewed the references recommended by the reviewer. In this case, although the term "connectivity" was not used, we consider that it is implicit in the relationship of the hydraulic and  the physical-chemical parameters.

Reviewer 3 Report

This paper presents an interesting case study focused on 2D water quality modelling in rivers, and in my opinion it deserves publication provided that the following minor changes are introduced.

1. The paper would benefit from a language revision, since there are several grammar mistakes throughout the paper.

2. The description of the model is detailed and clear, but it should not be included in the Introduction section, but rather in a specific section named “Numerical model” or something similar. Alternatively, and probably a better option, it can be included as a subsection in the “Materials and methods” section.

3. The numbering of the Materials and methods is wrong (number 1 should be number 2).

4. The authors report a computational time of 4 days. I guess such a long computational time is because they have probably not activated the IberPlus plugin, which enables the High Performance Computing implementation of Iber, performing parallelized computations in GPU (provided that the computer has an NVIDIA graphics card) or in CPU (provided that the computer has several threads). I guess this is the case because the authors do not cite the 2 papers in which the HPU implementation of Iber is presented, namely:

- Garcia-Feal, O., Cea, L., Gonzalez-Cao, J., Domínguez, J. M., & Gomez-Gesteira, M. (2020). IberWQ: A GPU accelerated tool for 2D water quality modeling in rivers and estuaries. Water, 12(2), 413.

- García-Feal, O., González-Cao, J., Gómez-Gesteira, M., Cea, L., Domínguez, J. M., & Formella, A. (2018). An accelerated tool for flood modelling based on Iber. Water10(10), 1459.

It would be interesting if the authors could do at least one simulation with the IberPlus implementation activated, in order to check the speed up in computational time. In García-Feal et al. (2020), speed-ups between 10 and 100 are reported, depending on the mesh size. This means that the computational time would probably be a couple of hours instead of 4 days. In any case, even if they do not perform this check, they should mention the HPC implementation of Iber and cite the previous references.

5. The calibration presented in section 2.1 shows very good results. However, the authors do not give many details about the calibration procedure (automatic or manually, and which criteria was used to quantify the comparison between observed and modelled)

6. The quality/resolution of Figures 6, 7 and 8 should be improved, otherwise it is difficult to read.

7. In Figure 6, the longitudinal profile of DO and EC seems coherent to me, but I don’t fully understand the oscillations in Temperature. Do the authors have any physical explanation for this behaviour?

8. I would start the conclusions with a general comment about the calibration and performance of Iber (second paragraph in the present manuscript).

Author Response

Point 1: The paper would benefit from a language revision, since there are several grammar mistakes throughout the paper.

Response 1: A native English speaker reviews the paper.

Point 2: The description of the model is detailed and clear, but it should not be included in the Introduction section, but rather in a specific section named “Numerical model” or something similar. Alternatively, and probably a better option, it can be included as a subsection in the “Materials and methods” section

Response 2: Modify In the text

Point 3: The numbering of the Materials and methods is wrong (number 1 should be number 2).

Response 3: Modify In the text

Point 4: The authors report a computational time of 4 days. I guess such a long computational time is because they have probably not activated the IberPlus plugin, which enables the High Performance Computing implementation of Iber, performing parallelized computations in GPU (provided that the computer has an NVIDIA graphics card) or in CPU (provided that the computer has several threads). I guess this is the case because the authors do not cite the 2 papers in which the HPU implementation of Iber is presented, namely:

- Garcia-Feal, O., Cea, L., Gonzalez-Cao, J., Domínguez, J. M., & Gomez-Gesteira, M. (2020). IberWQ: A GPU accelerated tool for 2D water quality modeling in rivers and estuaries. Water, 12(2), 413.

 - García-Feal, O., González-Cao, J., Gómez-Gesteira, M., Cea, L., Domínguez, J. M., & Formella, A. (2018). An accelerated tool for flood modelling based on Iber. Water10(10), 1459.

It would be interesting if the authors could do at least one simulation with the IberPlus implementation activated, in order to check the speed up in computational time. In García-Feal et al. (2020), speed-ups between 10 and 100 are reported, depending on the mesh size. This means that the computational time would probably be a couple of hours instead of 4 days. In any case, even if they do not perform this check, they should mention the HPC implementation of Iber and cite the previous references.

Response 4: It is considered in the last paragraph of conclusions. Also, references were included

Point 5: The calibration presented in section 2.1 shows very good results. However, the authors do not give many details about the calibration procedure (automatic or manually, and which criteria was used to quantify the comparison between observed and modelled)

Response 5:  It was considered in section 2.7 and 3.2

Point 6: The quality/resolution of Figures 6, 7 and 8 should be improved, otherwise it is difficult to read.

Response 6:  We change the figures

Point 7: In Figure 6, the longitudinal profile of DO and EC seems coherent to me, but I don’t fully understand the oscillations in Temperature. Do the authors have any physical explanation for this behaviour?

Response 7:  Temperature oscillations are in correlation with dissolved oxygen.

Point 8: I would start the conclusions with a general comment about the calibration and performance of Iber (second paragraph in the present manuscript).

Response 8:  Modify In the text

Round 2

Reviewer 1 Report

Manuscript is much improved 

Author Response

A native English speaker reviews the paper
